# Molecular Methods for Pathogenic Bacteria Detection and Recent Advances in Wastewater Analysis

Shuxin Zhang [1], Xuan Li [1], Jiangping Wu [1], Lachlan Coin [2], Jake O'Brien [3], Faisal Hai [1] and Guangming Jiang [1,4,*]

1   School of Civil, Mining and Environmental Engineering, University of Wollongong, Wollongong 2522, Australia; sz877@uowmail.edu.au (S.Z.); xuanl@uow.edu.au (X.L.); jw130@uowmail.edu.au (J.W.); faisal@uow.edu.au (F.H.)
2   Department of Microbiology and Immunology at the University of Melbourne, Melbourne 3010, Australia; lachlan.coin@unimelb.edu.au
3   Queensland Alliance for Environmental Health Sciences (QAEHS), University of Queensland, Woolloongabba 4102, Australia; j.obrien2@uq.edu.au
4   Illawarra Health and Medical Research Institute (IHMRI), University of Wollongong, Wollongong 2522, Australia
*   Correspondence: gjiang@uow.edu.au; Tel.: +61-02-4221-3792

**Abstract:** With increasing concerns about public health and the development of molecular techniques, new detection tools and the combination of existing approaches have increased the abilities of pathogenic bacteria monitoring by exploring new biomarkers, increasing the sensitivity and accuracy of detection, quantification, and analyzing various genes such as functional genes and antimicrobial resistance genes (ARG). Molecular methods are gradually emerging as the most popular detection approach for pathogens, in addition to the conventional culture-based plate enumeration methods. The analysis of pathogens in wastewater and the back-estimation of infections in the community, also known as wastewater-based epidemiology (WBE), is an emerging methodology and has a great potential to supplement current surveillance systems for the monitoring of infectious diseases and the early warning of outbreaks. However, as a complex matrix, wastewater largely challenges the analytical performance of molecular methods. This review synthesized the literature of typical pathogenic bacteria in wastewater, types of biomarkers, molecular methods for bacterial analysis, and their recent advances in wastewater analysis. The advantages and limitation of these molecular methods were evaluated, and their prospects in WBE were discussed to provide insight for future development.

**Keywords:** pathogenic bacteria; biomarkers; molecular methods; wastewater-based epidemiology; antimicrobial resistance

## 1. Introduction

Diseases induced by human pathogens are a major threat to public health worldwide. According to an estimation by the World Health Organization (WHO), 600 million people (almost 1 in 10 people) fall ill from eating contaminated food, and among them 420,000 people die every year, leading to the loss of 33 million healthy lives [1]. Unsafe food causes an annual loss of USD 110 billion in productivity and medical expenses in low- and middle-income countries. In addition, the ever-increasing waterborne diseases become a global burden, causing a financial loss of almost USD 12 billion and over 2.2 million deaths each year. Water and foodborne diseases overburden health care systems and affect global economies, thus hindering socio-economic development.

Among various waterborne and foodborne pathogens, pathogenic bacteria are the largest and most common group. Food and waterborne pathogenic bacteria, including *Escherichia coli* (*E. coli*), *Campylobacter* spp., (*C. jejuni* and *C. coli*), *Legionella* spp., *Salmonella* spp., and *Shigella* spp. are responsible for most infection cases, sometimes with severe and fatal outcomes. In recent years, although water and foodborne disease outbreaks have been declining with the increasing efforts in improving public health, the burden of

infectious water and foodborne diseases is still a pressing global issue [2,3]. Furthermore, the growing antimicrobial resistance (AMR) of pathogens threatens the effective prevention and treatment of an ever-increasing range of infections, reminding us of the urgency and importance of strengthening our capacity to monitor and prevent the increasing risk of these human pathogens.

The traditional detection method of pathogenic bacteria mainly relies on microbial culturing. Culture-based methods are low-cost, easy to operate, and highly standardized and thus are broadly used for the regulatory purposes of pathogenic bacteria monitoring such as enumerating FIB in bathing water [4]. However, the main limitations of these methods are the lack of differentiation between the target and other non-target endogenous microorganisms of the same samples, false negative/positive results, time and labor-consuming procedures, and the inability to detect viable but nonculturable (VBNC) cells [5]. Moreover, in the application of quantitative research, culture-based methods often underestimate the number of bacteria. This affects the quantification accuracy of targets and underestimates the prevalence of pathogens in the human community. In addition, VBNC can become viable and may cause disease and increase the public health risk [4]. Therefore, as a rapid analyzing tool with high accuracy and specificity, molecular methods have quickly become the mainstream detection technique of pathogenic bacteria.

According to the biological markers being used, molecular methods can be divided into two major groups, i.e., the nucleic acid targeting method and protein/antigen targeting method [6]. The nucleic acid targeting method includes fluorescence amplification-based methods, such as the polymerase chain reaction (PCR), quantitative or real-time PCR (qPCR), digital PCR (dPCR), deoxyribonucleic acid (DNA) microarray, fluorescence in situ hybridization (FISH), and molecular beacon, and sequencing-based methods such as pyrosequencing, Illumina sequencing, and nanopore sequencing. The protein and antigen targeting method includes a traditional antibody–antigen interaction method similar to immunological methods (lateral flow tests (LFTs)) and enzyme-linked immunosorbent assays (ELISA) [7]. Moreover, by combining the basic molecular detection approaches with a metal and paper platform, biosensor-based and paper-based devices have become a rapid, cheap, and portable on-site method for pathogenic bacteria detection [8–10].

Wastewater-based epidemiology (WBE) is a method to obtain qualitative and quantitative data on the chemical use/exposure and infectious cases of residents within a given wastewater catchment area based on the analysis of chemical compounds, pathogens, and certain biomarkers in raw sewage [11,12]. Sewage collected from a wastewater treatment plant (WWTP) or sewers can be regarded as the pooled urine and stool sample within a community and can be used to evaluate the health status of the whole community [11]. Studies based on wastewater have showed that sewage can reveal not only illicit drug use, diet, and lifestyles, but also disease outbreaks within a community [11–13]. In addition, different from clinical testing, wastewater analysis can include pathogens shed by asymptomatic and presymptomatic individuals. This makes it a powerful tool for the early warning and timely intervention of infectious disease [14–16]. Wastewater can be monitored for pathogenic and benign microbes through a variety of technologies. Traditional techniques usually include filtering, staining, and examining samples under a microscope. More sensitive and specific methods such as PCR-based methods and DNA sequencing have also been employed in the analysis of human pathogens in wastewater. However, the concentration of pathogenic bacteria is usually lower than indicator microorganisms and thus requires a highly sensitive detection method. The complex wastewater matrix often causes false-negative results because of the presence of various inhibitors. In addition, sample processing methods vary for different downstream analytical methodologies. Therefore, it is still challenging to utilize molecular detection methods in the accurate and quantitative detection of pathogenic bacteria in raw wastewater. This review focuses on the molecular analytical techniques for pathogenic bacteria in wastewater and summarizes recent advances of these approaches in wastewater-based epidemiology. The prospects of these approaches in wastewater analysis are discussed to provide insights for further application.

## 2. Pathogenic Bacteria in Wastewater

Human pathogens, causing serious infections and even death, are one of the leading threats to global public health. Various human pathogens can be grouped as bacteria (e.g., enterohemorrhagic *E. coli* (EHEC), *Campylobacter* spp., and *Salmonella* spp., etc.), viruses (e.g., influenza viruses, hepatitis virus, rotaviruses, and Norwalk viruses, etc.), protozoa (e.g., *Giardia lamblia* and *Cryptosporidium parvum*, etc.), and parasites (e.g., ascaris, Ancylostoma, etc.) [17]. Currently, there are approximately 538 species of pathogenic bacteria infecting human beings. This number is much higher than the overall number of other pathogens such as viruses (around 208 types), parasitic protozoa (around 57 species), and some fungi and helminths [17]. The bacteria listed in Table 1 are bacterial human pathogens detected in raw wastewaters, which are reported as the most common species that can cause global health concern (e.g., gastroenteritis or pneumonia). Most pathogens in wastewater are shed by human patients, although some might originate from other sources such as animals.

**Table 1.** Pathogenic bacteria detected in wastewater with high significance to public health.

| Pathogenic Bacteria | | Related Disease | Infectivity [a] | Persistence [b] | Density in WWTP Influent | Reference |
|---|---|---|---|---|---|---|
| Enteric pathogens | *E. coli* O157:H7 | Gastroenteritis | High | Moderate | $10^1$–$10^6$ CFU/100 mL | [18] |
| | *Campylobacter* spp. | Gastroenteritis | Moderate | Moderate | $10^2$–$10^5$ MPN/100 mL | [19] |
| | *Shigella* spp. | Shigellosis | High | Short | $10$–$10^7$ MPN/100 mL | [20,21] |
| | *Salmonella* spp. | Salmonellosis; Typhoid fever | Low | May multiply | $1$–$10^7$ MPN/100 mL | [20,22] |
| | *Clostridioides difficile* | Severe diarrhea and colitis | High | Long | - | [23,24] |
| Non-enteric pathogens | *Legionella* spp. | Acute respiratory illness, legionellosis | Moderate | May multiply | $10^7$–$10^{10}$ cells/100 mL | [25,26] |
| | *Mycobacterium* spp. | Pulmonary disease, skin infection | Low | May multiply | $10^5$ gene copies/100 mL | [27] |

Note: [a] [6]; [b] [17]; MPN: most probable number.

### 2.1. Enteric Pathogenic Bacteria

#### 2.1.1. *E. coli* O157:H7

The strain of *E. coli* that causes diarrhea is called diarrheagenic *E. coli*. Among several pathotypes of diarrheagenic *E. coli*, enterohemorrhagic *E. coli* (EHEC) is different because of its ability to produce Shiga toxin. Both of *E. coli* O157:H7 and other non-O157 STEC (Shiga toxin-producing *E. coli*) such as O26:H11, O111:H8, and O118:H16 can release Shiga toxins, but only O157:H7 genotypes can induce disease in humans. Others commonly reside in cattle without causing diseases [28]. *E. coli* O157:H7 is the predominant and most virulent serotype in the pathogenic subgroup of *E. coli*. It could cause not only diarrhea and hemorrhagic colitis, but also hemolytic uremic syndrome, which is a serious long-term complication, mainly affecting children, that leads to kidney failure and death. Virulence factors of *E. coli* O157:H7 include the type III secretion system, Shiga-like toxin 1 and 2, acid tolerance response system, hemolysin, and extracellular serine protease [29,30]. *E. coli* is widely used as the fecal contamination indicator, and the O157:H7 genotype is often employed as a model for pathogenic bacteria study in wastewater [18]. The density of *E. coli* O157:H7 in raw wastewater was found as $10$–$10^6$ CFU/100 mL [18].

#### 2.1.2. *Campylobacter* spp.

*Campylobacter* spp. is one of four major causes of diarrhea, and it is also regarded as the most common cause of human gastroenteritis worldwide. There are 13 pathogenic *Campylobacter* species known to be associated with human infections such as *C. jejuni*, *C. coli*, *C. lari*, *C. concisus*, *C. rectus*, *C. hyointestinalis*, *C. insulaenigrae*, *C. sputorum*, *C. helveticus*, *C. fetus*, *C. mucosalis*, *C. upsaliensis*, and *C. ureolyticus*. Among 17 species and six subspecies of *Campylobacter*, *C. jejuni* and *C. coli* are the most related to infections, accounting for 80–85% and 10–15% of total infections, respectively [31]. *C. jejuni* and *C. coli* are also the main species widely detected and isolated from wastewater [19,32,33]. Pathogenic Campylobacter is responsible for 400–500 million infections annually [34]. In Europe, nearly 230 thousand cases have been reported every year since 2015 [35]. Presumably, the infection dose of campylobacteriosis is very low, with 360 colony-forming units (CFU)

being adequate to cause the illness. *Campylobacter* spp. possess different virulence factors (VFs) related to motility, adhesion, invasion, toxin-activity, immune evasion, and iron-uptake [36]. VFs, such as the cadF gene and iam locus, are involved in different invasion steps [37,38]. Other VFs, such as the tripartite toxin encoded in the cdtA, cdtB, and cdtC genes [39], block the CDC2 kinase, inducing progressive cellular distension, which causes cell death [36]. Therefore, those genes are widely used in the diagnosis of *Campylobacter* spp. since thermotolerant *Campylobacter* spp. is difficult to culture [19,40,41].

### 2.1.3. *Salmonella* spp.

Salmonellosis is one of the most important zoonotic diseases caused by *Salmonella* spp. and transmits to humans through raw food products. A few serotypes, such as *S. typhimurium* and *S. enteritidis*, can cause human infection with poultry as the main host. Two foods that are most commonly associated with *Salmonella* infection are eggs and poultry meat [42]. According to the Centers for Disease Control and Prevention, from 2006 to 2017, *Salmonella* was responsible for about 53.4% of all foodborne disease outbreaks in the USA, and approximately 32.7% of these outbreaks were related to produce consumption [43]. Additionally, *S. Typhi* and *S. Paratyphi* are the main causes of typhoid fever and paratyphoid fever, respectively. Both are human-specific, Gram-negative, and human-restricted bacterial pathogens. Transmission can occur from person to person by eating contaminated food or water or by contact with an acute or chronic infected person [44]. Although a few culturing methods have been developed to isolate and culture those two *Salmonella* species in various samples, the cultivation is still a difficult task since they are fastidious microorganisms. The molecular approach is considered better for the detection and quantification of *Salmonella* spp. than the culture-based approach. Many protocols have been developed to target different genes or gene regions specific to *Salmonella* spp. The most popular gene targets are invA and flagellin genes (fliC-d for *Salmonella Typhi*, fliC-a for *Salmonella Paratyphi* A). Some tests are multiplexed to improve sensitivity and specificity [45–49]. Protein markers such as membrane vesicle protein PagC have also been employed in the detection of pathogenic *Salmonella* as a novel biomarker [50].

### 2.1.4. *Shigella* spp.

*Shigella* is a Gram-negative bacterium, a facultative anaerobe of the Enterobacteriaceae family. It is thought to be responsible for shigellosis or bacillary dysentery [51]. *Shigella* spp. includes four serotypes: *S. dysenteriae*, *S. flexneri*, *S. boydii*, and *S. sonnei*. Shigellosis is an invasive disease of the colon that is mainly caused by *S. sonnei* and *S. flexneri*. The main mode of transmission is fecal–oral infection, with an infection dose as low as 10 bacterial cells. The continuous transmission in humans must be passed from one person to another, as the bacterium does not survive long outside the body during the plankton-like phase [52]. About 165 million cases of *Shigella* disease are recorded worldwide each year, resulting in 1 million deaths, particularly in developing countries. It is reported that the phenotypic and genotypic characteristics of *Shigella* species are too close to be distinguished from diarrheagenic *E. coli*. This close genetic relationship between *Shigella* and *E. coli* leads to the widespread presence of virulence genes, making it difficult or even impossible to distinguish the virulence types of *Shigella* from *E. coli*. In recent years, in order to achieve rapid and reliable identification of the four *Shigella* species, a series of PCR-based methods has been developed by targeting various genes in plasmid DNA, including invasion plasmid antigen H (ipaH) [53], ial [54], virA [55], the she pathogenicity island (spi) [56] and tuf [57].

### 2.1.5. *Clostridioides difficile*

*Clostridioides difficile* (homotypic synonym *Clostridium difficile*, also known as *C. difficile*, or *C. diff*) is a kind of Gram-positive spore-forming bacterium [58]. Pathogenic *C. difficile* strains can induce diarrhea and life-threatening pseudomembranous colitis, often requiring antibiotic treatment. These strains are mainly identified by their ability to produce the enterotoxin A (TcdA) and/or the cytotoxin B (TcdB) [59]. *C. difficile*-associated diarrhea

(CDAD) is a very common nosocomial infection related with high morbidity and mortality, which imposes a huge financial burden to healthcare facilities [60]. In the last two decades, toxigenic *C. difficile* became one of the most important causes of hospital infections, with many infections leading to diarrhea and potentially fatal pseudomembranous colitis [61]. It has been recognized that community-acquired *C. difficile* infections are increasing among people with no apparent contact with healthcare facilities and without any known risk factors for *C. difficile* infection (CDI) [62]. Since *C. difficile* is a spore-forming bacterium, it is considered as an environmentally resistant pathogenic bacterium with the ability to prolong survival under environmental conditions. Therefore, *C. difficile* in the feces of both symptomatic and asymptomatic CDI patients can enter hospitals and domestic wastewater, which can be the possible approaches for CDI transmission within a community [63]. Moreover, AMR of *C. difficile* raises a major threat to the global health care system, not only because of the treatment of CDI, but also because it can be a reservoir of AMR genes to spread them to other pathogens [64]. These facts raise concerns of *C. difficile* infection and transmission and entail its surveillance based on WBE. Several studies have investigated the prevalence of *C. difficile* in wastewater by cultivation and PCR methods [59,65,66].

### 2.2. Non-Enteric Pathogenic Bacteria

Many of the pathogenic bacteria detected in wastewater are enteric in origin. However, a few of the pathogenic bacteria, which cause non-enteric diseases such as *Legionella* spp. and *Mycobacterium* spp., have also been detected in wastewater [67–69].

#### 2.2.1. *Legionella* spp.

Pneumonia caused by *Legionella* spp. is a life-threatening pulmonary infection that is mostly caused by *Legionella pneumophila* [70]. In addition, another 19 species have also been confirmed as human pathogens based on results isolated from clinical specimens [71]. Infections could be spread not only in communities, but also in hospitals. Moreover, Legionnaires' disease (LD) is clinically and radiologically indistinguishable from community-acquired pneumonia (CAP) caused by other bacterial pathogens [72]. For the treatment, *Legionella* spp. are unaffected by β-lactam antibiotics since they are obligatory intracellular bacteria. The treatment of infections thus requires a high dose of quinolones or macrolides [73]. Early diagnosis of LD is essential for the monitoring of outbreak and treatment in hospitals [74,75]. *Legionella* has been found in wastewater with concentrations up to $10^8$ CFU/L. A recent study found that exposure to aerosols dispersed from WWTPs caused LD in residents living near WWTPs during 2013–2018 in the Netherlands [26].

#### 2.2.2. *Mycobacterium* spp.

The *Mycobacterium* genus includes more than 170 species [76], of which at least two, *Mycobacterium tuberculosis* and *Mycobacterium leprosy*, are regarded as obligate human pathogens. Most others are opportunistic organisms that cause disease both in humans and animals when conditions are favorable. Generally, *Mycobacteria* are classified into two main groups, the genetically related *M. tuberculosis* complex (MTC) organisms and nontuberculous mycobacteria (NTM). The NTM are also known as environmental mycobacteria due to their widespread presence in soil and water [77]. Tuberculosis (TB) is a disease caused by infection with *M. tuberculosis*, which caused 1.4 million deaths in 2019. TB became one of the top ten causes of death and is the leading cause from a single infectious agent (ahead of HIV/AIDS). In addition, multidrug-resistant TB (MDR-TB) is also a public health crisis and a health security threat. In 2019, 206,030 patients with multidrug-resistant or rifampicin tuberculosis (MDR/RR-TB) were detected and reported globally, an increase of 10% from 186,883 in 2018 [78]. However, nontuberculous mycobacteria (NTM) have never been quantified in wastewaters before Radomski's study because of the inefficient analytical approaches [27]. More wastewater studies should be conducted for its environmental surveillance.

## 3. Molecular Methods for Pathogenic Bacteria Detection

### 3.1. Biomarkers of Pathogenic Bacteria

Biomarkers including nucleic acids, proteins, antigens, adenosine triphosphate (ATP), and metabolic products [10] are employed in the analysis of microorganisms. To differentiate microorganisms within one sample, nucleic acids (DNA/RNA), proteins, and antigens are usually selected as biomarkers because of their special physical and chemical characteristics within different pathogens. The detection of DNA/RNA is based on the specific hybridization and amplification of targets, thus enabling good specificity and accuracy. In case of pathogenic bacteria in wastewater, the most important biomarker is the pathogenic DNA or RNA residues from these bacteria. The biomarkers include the genus/species-specific genes, functional genes, and antimicrobial resistance genes [11]. Moreover, in the analysis of antimicrobial resistance, gene transfer is another significant point. Various mobile genetic elements, including plasmids, transposons, bacteriophages, integrons, and combinations of them, are notable nucleic acid targets for investigating the prevalence and spread of resistance genes in bacteria [79].

ATP assay, enzymatic activity tests, and metabolic products are mostly used to assess the activity of living cells. Due to the linearity between the total number of ATP and the total colony-forming units, the metabolically active cells could be directly quantified using the amount of ATP [80,81]. Ions and some organic acids are the metabolic products of microorganisms. These metabolites could be detected by electrochemical methods, thus were utilized to reflect the metabolic states of microorganisms [82].

Microbial surfaces contain a variety of proteins that are expressed by specific DNA/RNA in different pathogens. By screening these proteins using antibodies and nucleic acids, new biomarkers can be discovered, and pathogens can be specifically detected. Antigens are another kind of molecules on the cell surface of pathogens. They can be specifically bound to antibodies and induce immune responses of the host. Each type of pathogen carries one or more unique antigens on their surface, even within strains. It thus enables the specific identification of pathogens using antibodies [83]. Moreover, by analyzing the specific antigens of each strain, the subtypes of the strain can be determined.

Aptamers are single-stranded DNA or RNA oligonucleotides with high affinities and specificities that can bind a variety of targets, from single molecules to whole cells [84]. They can form diverse, complex secondary structures such as multi-branched loops and G-quadruplexes, which can specifically target the surface proteins of microorganisms or cells. In environmental monitoring, aptamers are superior to antibodies due to their chemical stability, easy chemical modification, relative ease of synthesis, and biocompatibility. With the systematic evolution of ligands by exponential enrichment (SELEX) method, many aptamers have been successfully employed to detect various pathogens in environmental samples [8,85,86].

### 3.2. Molecular Methods

For a long time, the culture and colony counting-based method has been the dominant method in the detection of pathogens (i.e., the 'gold standard'). It can assess live microbes or viable cells in samples. However, these methods may produce false-positive or false-negative results when evaluating highly aggregated microbial cells. Furthermore, not all microbial cultures can be grown under laboratory conditions. For example, a study of *Campylobacter* indicated that the culture-based method failed to correctly detect *Campylobacter* in 30% of positive patient stool samples compared to non-cultural methods, including PCR and enzyme immunoassay (EIA) [87]. Moreover, the culture methods are time and resource intensive, which are gradually replaced by more rapid and specific molecular methods. Therefore, in order to meet the requirements for reliable analysis of pathogenic bacteria, including high specificity, high sensitivity, good reproducibility, automation, and cost effectivity, molecular methods have gradually emerged to replace the dominant position of culture methods. In recent decades, various rapid, sensitive, and specific molecular methods have been developed. These molecular methods are discussed below and listed in Table 2.

Table 2. Molecular detection methods and example of applications in the analysis of pathogenic bacteria.

| | Molecular Method | | | Target Bacteria/Genes | Sample Type | Limit of Detection | References |
|---|---|---|---|---|---|---|---|
| Nucleic acid targeting methods | Polymerase chain reaction (PCR)-based method | Multiplex-PCR (mPCR) | | *Enteropathogens* | Wastewater | - | [88] |
| | | Single qPCR | | *Salmonella* | Salmonella Isolates | - | [89] |
| | | Multiplex qPCR | Taqman method | invA of *Salmonella* spp.; the paratose synthase (prt) gene, and the tyvelose epimerase (tyv) gene of group D and group A *Salmonella*, the *Salmonella*-differentiating fragment 1 (Sdf-1) sequence of *S. Enteritidis* | Environmental Samples | 1 copies/reaction [b] | [90] |
| | | | SYBR green | mcr-1 gene | Wastewater | 12 copies/reaction [b] | [91] |
| | | Microfluidic quantitative PCR | | Antibiotic resistance and heavy metal resistance genes | Wastewater | - | [92] |
| | | Droplet digital PCR (ddPCR) | | VBNC *E. coli* O157:H7/rfbE | Food | 5–6 copies/$\mu$L [b] | [93] |
| | | DNA Microarray | | *Salmonella enterica, Shigella flexneri, E. coli* O157:H7, and *Listeria monocytogenes* | Food | $10^2$ CFU/mL [a] | [94] |
| | | LAMP | | *V. parahaemolyticus* | Flatfish | 1 CFU/mL in buffer [b]; 10 CFU/g in fish sample [a] | [95] |
| | | FISH | | *Salmonella* | Minced lamb meat | 10 CFU/g [a] | [96] |
| | Sequencing | Pyrosequencing | | Bacterial communities | Sputum | - | [97] |
| | | Illumina technology | | 16S rRNA gene | Well-characterized bacterial reference sample | - | [98] |
| | | Oxford Nanopore Technologies | | | | | |
| | | Whole-genome sequencing (WGS) | | 381 different resistance genes | Wastewater | - | [99] |
| Immunology-based methods | Enzyme-linked immunosorbent assay (ELISA) | | | *S. enterica typhimurium.* | River water | $9.2 \times 10^3$ CFU/mL [a] | [100] |
| Biosensor-based methods | Cross-linking reaction between antibody and water-soluble cf-GQDs (carboxyl functionalized graphene quantum dots) | | | *E. coli* O157: H7 | Milk | $10^2$ CFU/mL [a, b] | [101] |
| Paper-based device | Origami paper-based device | | | *E. coli* | Bacteria culture | $10^3$ CFU/mL [b] within 35 min | [102] |
| | Paper-based ELISA | | | *E. coli* O157:H7 | Food | $1 \times 10^4$ CFU/mL [a, b] | [9] |

Note: [a] method LoD (based on sample volume/mass); [b] assay LoD (based on reaction or L); [a, b] both method and assay LoD.

### 3.2.1. Nucleic Acid Targeting Methods

Nucleic acid targeting methods are designed to detect the specific DNA/RNA of pathogens. It is achieved by the hybridization between target nucleic acid sequences and synthetic oligonucleotides. Thus, the species-specific gene of pathogens and virulence genes can be detected through nucleic acid targeting methods. They are usually fast, efficient, and do not require the culture of the pathogens. These methods include polymerase chain reaction (PCR)-based methods such as conventional PCR, real-time/quantitative PCR (qPCR), droplet digital PCR (ddPCR), multiplex PCR (mPCR), and other methods such as microarrays, loop-mediated isothermal amplification (LAMP), sequencing, and fluorescence in situ hybridization (FISH).

#### PCR-Based Method

PCR is the most common molecular-based technique for the detection and quantification of pathogens. PCR enables the detection of a single pathogenic bacteria by targeting specific DNA sequences [57]. Through this method, a small sample of a DNA sequence could be rapidly amplified into a large amount. This advantage enables the detection and quantification of a low amount of the target DNA sequence. It is thus widely used in the diagnosis of human pathogens. It significantly increases the sensitivity of detection of microorganisms at low concentrations in environmental samples [103,104]. PCR has already been utilized to the detection of a series of pathogenic bacteria such as *E. coli* and spores of *C. perfringens* [104,105].

Conventional PCR needs gel electrophoresis to detect the formation of PCR products. Real-time polymerase chain reaction, also called quantitative PCR, is the real-time detection of the PCR process during the amplification of the target DNA sequence. qPCR determines the PCR amplification by measuring specific dual-labeled probes or fluorescent signals emitted by inserting dyes. The fluorescence intensity reflects the amount of the template DNA. There is a linear relationship between the cycle threshold (Ct or Cq) value and the initial concentration of the template gene during the exponential period of PCR amplification. Thus, the concentration of target sequences could be calculated from a well-established standard curve to achieve an absolute quantification. Real-time quantitative PCR is widely used in the real-time detective and quantitative analysis of target DNA sequences with higher specificity and sensitivity than conventional PCR [106].

Two main fluorescence systems have been developed for qPCR, i.e., the SYBR green method and the TaqMan probes method. SYBR green is a fluorescent pigment that can bind double-stranded DNA (dsDNA). This non-sequence-specific pigment enhances the fluorescence signal when it binds to DNA double helix minor grooves, thus enabling the quantification of the targeting sequence. In contrast, TaqMan probe does not require the addition of fluorescent pigment. The template-specific TaqMan probe further improves the specificity of qPCR by increasing primer specificity. For each amplification of a specific target, one molecule of fluorescent dye is released. The instrument detects the fluorescence produced by specific amplification, which is not impacted by non-specific amplification. This ensures the high specificity of the qPCR detection. There are many reporter–quencher sets with different wavelengths, which can be labeled with the TaqMan probe. This enables the TaqMan method to be able to detect multiple PCR reactions in the same tube, leading to reduced cost and improved efficiency and accuracy. It can also avoid the influence of different fluorescent dyes on the PCR reaction.

The mPCR is a faster detection methodology than simplex PCR, which can detect multiple gene targets simultaneously. Fan et al. (2008) reported one mPCR assay to achieve the simultaneous detection of various human pathogens in a single tube, with the detection sensitivities between 10 to $10^2$ CFU/100 mL in seawater. To differentiate the pathogenic and commensal *E. coli* in clinical and water samples, an mPCR assay was developed to detect the occurrence of 11 virulence genes in *E. coli* [107]. Recently, the presence of enteropathogens in sewage was investigated by using the commercially available FilmArray® mPCR system [88].

Compared with simplex PCR, mPCR provides faster detection by simultaneously amplifying multiple gene targets. It can also differentiate closely related pathogenic bacteria.

Digital PCR is a biotechnology improvement on conventional PCR and can be used to directly amplify and quantify DNA, cDNA, or RNA. Droplet digital polymerase chain reaction (ddPCR) is a kind of dPCR technique that is emerging as a powerful analytical tool for absolute quantification. Similar to qPCR, ddPCR also utilizes Taq polymerase to amplify a target DNA sequence in a standard qPCR assay. The differences are that ddPCR separates the whole qPCR reaction into thousands of individual reactions before amplification, and ddPCR collects data at the reaction end point. These differences provide ddPCR many advantages, such as the direct and independent quantification of target DNA without standard curves and more precise and reproducible data than conventional qPCR, especially when PCR inhibition is present [108–110]. In comparison to qPCR, ddPCR shows better performance in detecting low concentrations of target genes in environmental samples. Moreover, it has the potential to reduce the effect of qPCR inhibitors, although its application to complex environmental samples needs further optimization [111]. However, qPCR is more reliable in detecting higher concentrations ($2 \times 10^5$ or $2 \times 10^4$ gene copies/PCR), since ddPCR displays higher variability and less precision in these concentration ranges [112]. A recent study shows that, in addition to being faster, the ddPCR method exhibited higher sensitivity with a limitation of $10^{-5}$ ng/µL for genomic DNA templates and $10^{-1}$ CFU/mL for *Shigella* bacteria culture, when compared to PCR and qPCR [113].

Nowadays, with the increasing availability of sequencing data, it is theoretically possible to design qPCR assays for every microorganism [106]. The qPCR method has many benefits over other techniques. Firstly, the quantitative data produced by qPCR method could reach an accurate dynamic range of 7–8 log orders of magnitude without requiring post-amplification manipulation. Secondly, although the sensitivity of qPCR is varied towards different samples and might be inhibited by inhibitors, it has been reported to have higher sensitivity than many other molecular methods [114]. Theoretically, it is high enough to detect a single copy of a transcript. Studies have shown that qPCR is reliable and sufficient for the quantitative detection of various pathogens such as *E. coli* O157:H7 [115] and *Campylobacter* spp. [116] The qPCR method has been applied to detect and monitor the occurrence and concentrations of pathogens in drinking water sources [117], water and wastewater treatment plants [118,119], and recreational beaches [120]. The use of qPCR in water analysis enables quick microbial risk assessment, which may lead to immediate remedial actions.

However, for the detection and quantification of waterborne bacteria with low abundance, high requirements are demanded for qPCR, such as high accuracy, low limit of detection (LoD) and quantification (LoQ), and the ability to distinguish dead and viable cells. Studies have reported that it is possible to detect viable cells by detecting messenger RNA (mRNA), since it only exists in viable organisms [7,121]. However, not all mRNAs are present in all life phases of an organism, thus the target mRNA should be carefully chosen for viable organism detection. In addition, rRNA-based RT-qPCR assays were also confirmed to have a better association with the active bacterial populations in surface water samples than rDNA-based assays [122–124]. Furthermore, multiplex real-time PCR has been reported to be a valuable technique for the identification of viruses [125], bacteria [126], and parasites [127]. However, due to the limitation of instruments and the fluorescence groups, only four targets could be detected at the same time in TaqMan methods. This limitation prohibits its application in profiling microbiome communities in complex samples.

DNA Microarrays

DNA microarrays, also known as DNA arrays, are commonly known as gene chips. It is a special piece of glass or silicon chip with a DNA microarray, which places thousands or tens of thousands of nucleic acid probes on an area of several square centimeters [128]. DNA, complementary DNA (cDNA), and RNA in the sample are detected by fluorescence

or electric signal after being combined with the probes. DNA microarrays enable the hybridization-based detection of multiple targets in a single experiment, which makes it suitable for the analysis of massive targets. Using a high-throughput DNA microarray assay, a study investigated the prevalence of 941 pathogenic bacteria in groundwater and differentiated their sources of origin [129]. In general, DNA microarray allows for the simultaneous detection of multiple pathogenic bacteria. It is thus a fast and reliable diagnostic method in analyzing large numbers of clinical/environmental samples. However, the complicated probe design work, the reliability of the microarray data, and the clinical applicability of the early results have been criticized [130]. The criticism and intensified competition from other technologies, such as next-generation sequencing (NGS), have hampered the growth of microarray-based testing in the molecular diagnostics market [131].

Loop-Mediated Isothermal Amplification (LAMP)

LAMP is an isothermal nucleic acid amplification technique. It has been utilized for the alternative detection of certain diseases because of its low cost. At present, LAMP has been applied to the identification and quantification of pathogenic bacteria with significant advantages in sensitivity, specificity, and rapidity [95,132]. Since LAMP requires four primers specifically designed for six different regions of the target, any incomplete matching of the primers will theoretically lead to the phenomenon of non-reaction and non-specific amplification. In addition, the LAMP method was confirmed to be 10–100 times more sensitive than PCR detection [133], with a detection limit of 10 copies or less in the template for one reaction. Furthermore, it can directly detect pathogenic microorganisms in diseased tissue, thus avoiding the tedious cultivation and nucleic acid extraction step [134]. Finally, and most importantly, the result of the reaction can be judged with naked eyes by demonstrating the absence of the target gene with the production of white precipitate of magnesium pyrophosphate. It is more difficult to design specific primers for LAMP than PCR (because LAMP requires 4–6 primers and PCR requires only two). A software tool named PrimerExplorer is available to help the primer design for LAMP (http://primerexplorer.jp/e/). Therefore, as a rapid detection method without the need of any equipment, LAMP shows great potential in the rapid diagnosis of human pathogens in various samples.

Fluorescent in Situ Hybridization (FISH)

FISH is a cytogenetic technique used to detect and locate nucleic acids in cells or sample matrices. Fluorescently labeled nucleic acid probes hybridize only with highly similar nucleic acids and can be used to locate genes on chromosomes or to label ribosomal RNA in different taxonomic bacteria or archaea in molecular ecology. FISH could be employed in the enumeration of particular microbial populations [135]. Compared to PCR, FISH is more suitable for complex matrices because of its lesser sensitivity to inhibitory substances. However, a major limitation of FISH is the small number of phylogenetically distinct targets that can be detected at the same time. A recent study developed a multi-FISH method that uses eight fluorophores, which is highly suitable for investigating the structure and function of microbial communities in different samples [96]. Furthermore, FISH has been used to discover emerging human pathogens in water, wastewater, and sludge, to produce quantitative descriptions of the microbial community in wastewater and activated sludge [136,137] and to investigate survival and infection mechanisms at the cellular level. However, this method is still partly based on cell culture.

Sequencing

Sequencing is the process of determining the sequence of nucleotides in a section of DNA. It includes any method or technique used to determine the order of the four bases: adenine, guanine, cytosine, and thymine (or uracil for RNA). In 1977, DNA sequencing technology was firstly developed by Frederick Sanger based on the chain-termination method (also known as Sanger sequencing). In the early stage, DNA sequencing was

employed for small genomes such as viruses and organelles. Complete sequencing of a bacterium genome was not feasible because of the economic and technical limitations. Later, with the emergence of the shotgun method developed by Sanger et al., whole genome sequencing of bacteria was achieved. The shotgun method is considered the gold standard, and whole genome sequencing of many bacteria has been carried out using this method over years [138].

Next-generation sequencing (NGS), also known as high-throughput sequencing, is the overall term used to describe several different modern sequencing pathways. These technologies allow DNA and RNA to be sequenced faster and at lower cost than Sanger sequencing, which was previously used, thus revolutionizing genomics and molecular biology research [139]. NGS technologies include Illumina (Solexa) sequencing, Roche 454 sequencing, and proton/PGM sequencing. The NGS technologies achieve high throughput and reduced cost by using massively parallel analysis, which allows 300 Gb of DNA to be read in a single run on a single chip. The four main advantages of NGS over classical Sanger sequencing are: (i) NGS needs significantly less DNA, as it can obtain a sequence from a single strand; (ii) NGS is significantly quicker than Sanger sequencing by combining the two separate processes of Sanger sequencing, i.e., chemical reaction and signal detection, in some versions of NGS; (iii) NGS is more cost-effective due to reduced time, manpower, and reagents; (iv) repeats in NGS caused by many short overlapping reads lead to a more accurate and reliable sequence, even though individual reads are less accurate. These advantages enable a great potential of NGS in the application of environmental research. NGS is capable of producing large numbers of reads at exceptionally high coverages throughout the genome with dramatically reduced cost through the massively paralleled approach. However, NGS requires the amplification of DNA molecules, which introduces random errors in the DNA synthesis. The amplified DNA strands would become progressively out-of-sync, which means the signal quality deteriorates as the read length grows. Therefore, long DNA molecules must be broken up into smaller pieces to maintain the quality of the reading, leading to a critical limitation of second-generation sequencing [140].

To solve the limitation, third-generation sequencing (TGS) technologies were developed to produce substantially longer reads than NGS by the direct sequencing of single DNA molecules. Nanopore sequencing (Oxford Nanopore Technologies, Oxford, UK) is a representative TGS approach for the sequencing of biopolymers, specifically polynucleotides in the form of DNA/RNA. Through nanopore sequencing, individual molecules of a DNA/RNA can be sequenced without PCR amplification or chemical labeling of the sample. Nanopore sequencing has a great potential in providing relatively low-cost genotyping, high mobility for testing, and the ability to rapidly process samples and display results in real time [141]. Applications of this method in the rapid identification of viral pathogens [142], plant genome sequencing [143], monitoring of antibiotic resistance [144], and haplotyping [145] has been reported. One major limitation of nanopore sequencing is its high raw read error rate, which remains between 5% and 15% despite recent improvements in nanopore chemistry and computational tools [138]. However, according to the latest updates from Nanopore technologies (accessed on 15 October 2021), an accuracy of 98.3% could be achieved through the production software MinKNOW 4.3 ("Super-accuracy" basecalling model) and Guppy 5 (https://nanoporetech.com/accuracy). In addition, the quality of the sequencing result is affected by library quality and the presence of sequencing inhibitors. Although more efforts are needed to improve the quality of Nanopore sequencing results, studies have confirmed that it has better bacterial identification performance in complex samples than traditional Illumina platforms [146]. Winand et al. compared the bacterial identification performance of second (Illumina) and third-generation sequencing technologies (Nanopore sequencing, Oxford Nanopore Technologies, Oxford, UK) by targeting the 16S rRNA gene. The results revealed that both techniques provide reliable identification of bacterial genera but may mislead the identification of bacterial species and constitute viable alternatives to Sanger sequencing for rapid analysis of mixed samples without any culture steps [98].

### 3.2.2. Immunology-Based Methods

Immunological methods are based on the specific interaction between antibodies and antigens. These methods include enzyme-linked immunosorbent assays (ELISA), immunofluorescence assays (IFA), and serum neutralization tests (SNTs) [6]. For immunology-based methods, specific fluorochrome labeled antibodies are used to capture targeted antigens, which serves as the enumeration of fluorescently labeled cells by detecting the fluorescence signal using microscopy or flow cytometry. However, the biomarkers of these methods should be carefully chosen to achieve specific detection at different classification levels including genus, species, and serotypes. The detection of *S. typhimurium* on an immunochromatographic strip was reported by Park et al. (2010). This study achieved the quantitative detection of *S. typhimurium* in the range of $9.2 \times 10^3$ to $9.2 \times 10^6$ CFU/mL in river samples within 20 min [100].

Although these methods can specifically detect targeted bacteria and their toxins and can be multiplexed for multiple samples, they are still limited by false-negative results and cross-reactions with similar antigens. False-negative results are a serious problem and often happen to classical methods. They can be induced by various inhibitory compounds and matrices of different types of samples, which vary largely and thus might cause different effects on the analytical performance of different detection methods. In addition, cross-reaction is another big problem for immunology-based methods. In one study, a monoclonal antibody was used for specifically detecting *E. coli* O157:H7 lipopolysaccharide (LPS1) [147]. The cross-reactivity with other bacteria happened due to the presence of a constituent sugar of LPS. One recent method comparison study for *C. difficile* surveillance in Switzerland showed that, compared to the PCR method, enzyme immunoassay led to more false-negative results of human stool samples [148]. Immunological methods usually require pre-enrichment to expose surface antigens, which leads to extended detection time. Moreover, due to their lower sensitivity than other molecular methods, immunology-based methods were less employed in the direct detection of pathogenic bacteria in wastewater samples [9].

### 3.2.3. Biosensor-Based Methods

A biosensor is an analytical platform composed of two elements: a bio-receptor and a transducer. Bio-receptors are responsible for recognizing the targets such as enzymes, proteins, nucleic acids, and cell receptors. After recognition, the transducer converts the biological interactions into electrical signals that can be measured (e.g., optical, electrochemical, or magnetic). Biosensors provide a rapid, real-time, on-site, and multiple detection of bacteria. Optical biosensors are selective, sensitive, and can be used for real-time monitoring of toxins, drugs, and pathogens in wastewater [149]. For example, by applying a fluorescently labeled specific aptamer, Yildirim et al. developed a portable optical biosensor for the indirect sensing of an *E. coli* O157:H7 strain in wastewater [150]. Surface-enhanced Raman scattering (SERS) pathogen biosensors, with noble metal nanoparticles (e.g., silver and gold) as an impressive substrate, have become an attractive research field. The colorimetric changes induced by the hybridization between single-stranded DNA probes modified by gold nanoparticles and their complementary DNA can avoid the requirement of expensive and complex fluorescent labeling [151]. Gold nanoparticles are widely used in biosensor instruments, especially for dark water samples [152]. Another notable biosensor for *E. coli* O157:H7 used carboxyl functionalized graphene quantum dots (cf-GQDs) to label a specific antibody [101]. This sensor can specifically recognize *E. coli* O157:H7 from different sources, such as water and food, with the minimum detection limit of 100 CFU/mL. However, the sensitivity to changes in pH, mass, and temperature are some of the challenges that must be addressed in using biosensors for bacterial pathogens in wastewater [153].

### 3.2.4. Paper-Based Device

A paper-based device is a small analytical tool that is printed by a wax printer and has different functional areas. It can integrate all the processes required for nucleic acid detection (enrichment, extraction, amplification, and visual detection) into a cheap paper

material [154]. The whole detection process can be completed by folding paper-based devices in different ways and in different sections, which overcomes the limitation of PCR tests. Paper-based device can achieve multichannel, sensitive detection, comparable to PCR detection, and provide a high-quality, rapid, and accurate diagnosis of pathogens [155]. Moreover, paper-based devices are easy to stack, store, and transport because they are thin, lightweight, and of different thicknesses [156]. Similar to biosensors, paper-based devices can also be used to target a variety of biomarkers, including nucleic acids, proteins, antigens, and chemicals [9,102,155,157]. By integrating various molecular detection methods, paper-based devices have emerged as a powerful platform for the fast diagnosis of pathogens and the determination of infection transmission [9,13,102]. However, the shelf life of paper-based device limits its further applications. Some paper-based devices contain reagents with a short shelf life, such as enzymes, and thus need to be stored in a refrigerator or freezer to maintain the activity of the reagents [158,159]. Studies about the shelf life of enzymes on paper-based devices have yielded some promising results and proven techniques, although further research is still needed. Furthermore, the analytical performance of paper-based devices is also limited by features of paper, including the paper fibers and pattern. Moreover, the coffee ring effect of paper-based devices can cause non-uniform distributions of detection reagents and samples, thus affecting the detection accuracy [158]. To overcome these limitations, future efforts should be put into developing more uniform papers and modifying the size and shape of the test zone.

## 4. Recent Advances of Molecular Methods for Pathogenic Bacteria in Wastewater

Current estimates of the burden of infectious diseases are often based on severe cases requiring hospitalization, which fails to cover asymptomatic patients. The emerging wastewater-based epidemiology (WBE) is based on the analysis of biomarkers in raw wastewater, which is then used to back-estimate the status of public health. Wastewater is a complex mixture of chemicals and microorganisms in water. It contains chemical and biological information directly discharged from our bodies. From a surveillance point of view, urban wastewater is an attractive resource, since it provides sampling material within a large and mostly healthy population.

The WBE approach was first outlined as a potential tool to evaluate the use of illicit drugs and misused therapeutic drugs within a community [160]. To date, WBE has become an important tool for estimating illicit and licit drug consumption by detecting and quantifying unchanged drugs and their human-specific metabolites in wastewater [12]. WBE studies also showed that wastewater can reveal not only illicit drug use and diet, but also infectious disease risk within a community [11,161,162]. Many studies have validated the feasibility of various molecular methods in wastewater. Sensitive and specific methods such as PCR, real-time PCR, and DNA sequencing have been employed in the analysis of wastewater to achieve the fast detection and accurate quantification of human pathogens [90,163,164] or the evaluation of community structure and antimicrobial resistance level [165–167].

### 4.1. Sample Processing and DNA/RNA Extraction Methods

Wastewater components, including fats, proteins, humic acids, and fulvic acids, can lead to problems in the downstream analysis (molecular detection). Wastewater sample processing is a key step for the detection of pathogenic bacteria by separating, concentrating, extracting, and purifying biomarkers for further analysis. The availability of different commercial DNA/RNA extraction kits showed variable efficiency when extracting samples such as wastewater and sediment. Table 3 lists several comparison studies of different sample storage, pre-treatment, and DNA/RNA extraction methods for various downstream analyses.

**Table 3.** Comparison studies of DNA/RNA extraction methods for various downstream molecular methods of wastewater and sediment samples.

| Downstream Analysis | Targets | Best/Limited Fragment Length | Suggested Extraction Kits/Methods | Storage and Pretreatment of Samples | Recovery Efficiency | Sample Type | Reference |
|---|---|---|---|---|---|---|---|
| **PCR-based method** | Lambda DNA | - | FastDNA Spin Kit for Soil | Stored at −70 °C | 15.5% to 43.3% | Sediment | [168] |
| | *Ancylostoma caninum* ova | - | MO Bio Power Max® Soil DNA Extraction Kit (MO BIO Laboratories Inc, Carlsbad, CA USA); Filtration | Stored at 4 °C in the dark | Treated wastewater: 39–50% Raw wastewater: 7.1–12% | Wastewater | [169] |
| **Microarray** | 16S Rdna, cpn60, and wecE | Detection sensitivity is optimal when DNA targets > 500 bp | Bead beating separation and ammonium acetate purification | Centrifuged at 3000× *g* for 16 min at room temperature; stored at −20 °C | 81 µg DNA/mL | Wastewater | [170] |
| **NGS** | ARG | 150 bp (Limitation of the sequencing length) | FastDNA Spin Kit for Soil | Ethanol fixation (50%); filter-concentrated using 0.22-µm mixed cellulose ester filters; stored at –20 °C | 10.3 ± 3.6 µg/sample | Wastewater | [171] |
| | 16S rRNA amplicons | | Qiagen Mini Kit and MO Bio PowerSoil Kit | Centrifuged at 10,000× *g* for 5 min to pellet; filtered through 0.22 µm cellulose nitrate membrane filters | - | Wastewater | [172] |

Mumy and Findlay developed an external DNA recovery standard for sediments by comparing the performance of three commercial kits (UltraClean™Soil DNA, FastDNA®SPIN®, and Soil Master™DNA Extraction) [168]. The results indicated that the FastDNA®SPIN® kit has the highest recovery rate and makes it possible to collect additional DNA by cleaning beaded sediments. Gyawali et al. investigated six rapid DNA extraction methods for recovering *Ancylostoma caninum ova* DNA from wastewater and reported that the filtration method recovered higher DNA concentrations in both treated and raw wastewater than centrifugation, hollow fiber ultrafiltration (HFUF), and flotation [169]. A comparative study about the relative effectiveness of 10 different bacterial DNA extraction methods for wastewater samples showed that only a few could achieve satisfactory results when applied to bacterial pathogens [170]. The method of combined bead beating separation and ammonium acetate purification was suggested as the most suitable approach for bacterial DNA extraction from wastewater prior to specific microbial detection using microarray hybridization technology. Li et al. compared the ARG sequencing analysis results of three DNA extraction methods [171]. It was found that ARGs captured by the FastDNA SPIN Kit for Soil had the highest DNA yield, purity, and diversity. Moreover, no discernable effects were found on ARG profiles with fixation in ethanol, deep-freezing, and overseas transportation of samples compared with fresh samples. Another comparative study indicated that the DNA Mini kit and PowerSoil kit produce the most consistent sequencing results in water and wastewater [172]. Collectively speaking, the performance of DNA/RNA extraction methods of wastewater varies, and it is essential to develop standard DNA/RNA extraction methods for different downstream analysis methods to achieve high recovery and quality of the nucleic acids of various pathogens in wastewater.

### 4.2. Detection and Quantification of Pathogenic Bacteria

The low concentration of targeted pathogenic bacteria in wastewater samples brings difficulties to their detection, which entails high sensitivity and repeatability. As a complex matrix, wastewater contains various inhibitors for a number of molecular methods, which is a significant challenge for their application in wastewater analysis. In addition, from a disease surveillance perspective, wastewater samples should be analyzed quickly enough to provide an early warning. Several molecular approaches have showed great potential in rapid analysis. Table 4 critically compares various detection methods used for wastewater samples in the last five years.

**Table 4.** Applications of molecular methods in the analysis of pathogenic bacteria in wastewater.

| Detection Method | Cultivation | DNA/RNA Extraction | Target Pathogen | Biomarker | Sample Type | Limit of Detection (LoD) | References |
|---|---|---|---|---|---|---|---|
| **PCR** | Yes | Yes | *Campylobacter* spp., *C. jejuni*, *C. coli* | 16S rRNA, mapA, ceuE | Wastewater | 2 CFU/100 mL | [173] |
| | | | *E. coli* O157:H7 | stx2 | | | |
| | | | *S. typhimurium* | stx1 | | | |
| **Real-time PCR** | No | Yes | *E. coli* | uidA gene | Wastewater | 10 gc/reaction (standard curve) | [174] |
| | No | | *Simkania negevensis* | 16S rRNA gene | Wastewater | 5 gc/reaction (standard curve) | [175] |
| | No | Yes | *S. enterica* serovar *Typhi* | stgA | Wastewater | 0.05–0.005 CFU/mL of seeded wastewater | [176] |
| | | | *S. enterica* serovar *Paratyphi A* | SSPAI | | | |
| **Droplet digital PCR (ddPCR)** | Yes | Yes | Shiga toxin-producing *E. coli* | stx2 | River water, wastewater | 6 gc/reaction of standard curve; 32 copies/100 mL in river water | [177] |
| **qPCR array** | No | Yes | All | 285 ARGs and nine transposase genes | Wastewater | - | [166] |
| | No | Yes | All | 229 ARGs and 25 mobile genetic elements | Wastewater | - | [178] |
| **Microfluidic qPCR** | No | Yes | All | ARGs, heavy metal resistance genes, genes encoding the integrase, and 16S rRNA genes | Wastewater, drinking water | - | [92] |
| **LAMP** | Yes | Yes | *Listeria monocytogenes* | lmo0753 gene | Wastewater | 65 fg/µL of DNA and 38 CFU/mL in cell culture | [179] |
| **FISH** | Yes | No | *Salmonella* spp. | 23S rRNA | Wastewater | $10^2$, 10, and 1 CFU/mL for 0 h, 6 h, and 24 h of enrichment in Rappaport-Vassiliadis broth, respectively. | [137] |
| **Biosensor-based device** | No | No | ARG | mecA gene | ARG-spiked wastewater effluent | 70 pM ($4 \times 10^7$ gc/µL) by bootstrapping | [180] |
| **Paper-based device** | No | No | *Salmonella typhimurium* | fimA | Spiked wastewater | $10^2$ CFU/mL | [181] |
| | Yes | No | β-lactamase-expressing bacteria | β-lactamase | Wastewater | $3.8 \times 10^6$ CFU/mL | [157] |
| **Sequencing** | No | Yes | All | ARGs | Sludge | - | [182] |
| | No | Yes | Shotgun metagenomic for microbial community analysis and pathogen detection | | Wastewater | Detected 87 pathogenic/opportunistic Bacteria, with most having <1% abundance. | [183] |
| | Yes | Yes | Nanopore and Illumina metagenomics analysis for mobile antibiotic resistome | | Wastewater | - | [25] |
| | No | Yes | Full-length 16S rRNA | | Wastewater | - | [184] |
| | No | Yes | 16S-rRNA | | Wastewater | - | [185] |

So far, the PCR-based method is the most popular molecular approach for specific pathogenic bacteria detection and quantification (relative and absolute quantification) in wastewater because of its high sensitivity, specificity, and the low cost compared to the sequencing method. Plenty of primer-probe sets targeting various human pathogenic bacteria have been developed. Those primer-probe sets showed high specificity, sensitivity, and efficiency for pathogenic bacteria detection at genus and species levels. A series of PCR-based methods with or without bacteria isolation and cultivation procedures has been reported for the fast detection and quantification of pathogenic bacteria in wastewater samples [173–175]. Recently developed primer-probe sets for the bacterial pathogen detection of various PCR-based methods are listed in Table 5. Some of them have been confirmed to have good performance in wastewater samples and thus could be potentially used in wastewater analysis, although further feasibility studies should be conducted. Among the PCR-based methods, ddPCR has been increasingly reported as having better analysis performance than traditional PCR-based methods in wastewater [186]. However, some studies have still claimed that the sensitivity of qPCR is superior to ddPCR in some circumstances (e.g., high targeted gene concentration), and their performance might vary with different assays [187].

**Table 5.** PCR primer-probe sets available for the detection and quantification of typical pathogenic bacteria in wastewater and other samples.

| Pathogenic Bacteria | Available PCR Primers and Probes (5′—3′) | Sensitivity | Reference |
|---|---|---|---|
| *E. coli* O157:H7 | RFBEO157-F GGATGACAAATATCTGCGCTGC<br>RFBEO157-R GGTGATTCCTTAATTCCTCTCTTTCC<br>RFBEO157-P HEX-TACAAGTCCACAAGGAAAG-BHQ1 | 1 CFU/g of seeded meat products after 4 h enrichment period at 37 °C | [188] |
| | Rfb-F GTGTCCATTTATACGGACATCCATG<br>Rfb-R CCTATAACGTCATGCCAATATTGCC | 2 CFU/100 mL of raw sewage | [173] |
| *Campylobacter* spp. | 16S-F CCTGAMGCAGCAACGCC<br>16S-R CGGAGTTAGCCGGTGCTTATT<br>16S-P FAM-CTCCGAAAAGTGTCATCCT –MGB | 3.2 gene copies/reaction | [19,40] |
| *C. jejuni* | hipO-F CTTGCGGTCATGCTGGACATAC<br>hipO-R AGCACCACCCAAACCCTCTTCA<br>hipO-P VIC-ATTGCTTGCTGCAAAGT- MGB | $2.0 \times 10^2$ CFU/g of feces | [31] |
| *C. coli* | glyA-F AAACCAAAGCTTATCGTGTGC<br>glyA-R AGTGCAGCAATGTGTGCAATG<br>glyA-P FAM-CAACTTCATCCGCAAT- MGB | $2.5 \times 10^2$ CFU/g of feces | |
| *C. lari* | glyA-F CAGGCTTGGTTGTAGCAGGTG<br>glyA-R ACCCCTTGGACCTCTTAAAGTTTT<br>glyA-P TET-CATCCTAGTCCATTCCCTTATGCTC ATGTT-TAMRA | 2.1 gene copies/reaction | [19] |
| *Shigella* spp. | ipaH-F CGCAATACCTCCGGATTCC<br>ipaH-R TCCGCAGAGGCACTGAGTT<br>ipaH-P FAM- AACAGGTCGCTGCATGGCTGGAA-BHQ1 | $10^{-5}$ ng/µL for genomic DNA templates, $10^{-1}$ CFU/mL for *Shigella* bacteria culture | [113] |
| *Salmonella* spp. | invA-F AACGTGTTTCCGTGCGTAAT<br>invA-R TCCATCAAATTAGCGGAGGC<br>invA-P TGGAAGCGCTCGCATTGTGG | 9–15 CFU/25 g food sample | [189] |
| *S. Typhi* | stgA-F TATCGGCAACCCTGCTAATG<br>stgA-R TATCCCGCCGG TTGTAAAT<br>stgA-P FAM-CCATTACAG CATCTGGCGTAGCGA-BHQ1 | 0.05–0.005 CFU/mL of wastewater | [176] |
| *S. enterica* serovar *Paratyphi A* | SSPAI-F ACCATCCGCAGGACAAATC<br>SSPAI-R GGGAGATTACTGATGGAGAGATTAC<br>SSPAI-P Cy5-AGAGTGCAAGTGGAGTGCCTCAAA-BHQ2 | | |
| *C. difficile* | tpi-F AAAGAAGCTACTAAGGGTACAAA<br>tpi-R CATAATATTGGGTCTATTCCTAC | For simultaneous identification and toxigenic type characterization (fecal and urban water samples) | [59,66] |
| | tcdB-F GGAAAAGAGAATGGTTTTATTAA<br>tcdB-R ATCTTTAGTTATAACTTTGACATCTTT | | |
| | tcdA-F AGATTCCTATATTTACATGACAATAT<br>tcdA-R GTATCAGGCATAAAGTAATATACTTT | | |
| *Legionella* spp. | PanLeg-F GGCGACCTGGCTTC<br>PanLeg-R1 GGTCATCGTTTGCATTTATATTTA<br>PanLeg-P1 FAM-ACGTGGGTTGCAA-MGBNFQ | | |
| *L. pneumophila* | Lp-F TTGTCTTATAGCATTGGTGCCG<br>Lp-R CCAATTGAGCGCCACTCATAG<br>Lp-P Quasar670-CGGAAGCAATGGCTAAAGGCATGCA-BHQ3 | 5 genome units (GU)/reaction with water sample | [190] |
| *L. pneumophila sg1* | Lp1-F TGCCTCTGGCTTTGCAGTTA<br>Lp1-R CACACAGGCACAGCAGAAACA<br>Lp1-P VIC-TTTATTACTCCACTCCAGCGAT-MGBNFQ | | |
| *Mycobacterium* spp. | 16S rRNA-F: ATGCACCACCTGCACACAGG<br>16S rRNA-R: GGTGGTTTGTCGCGTTGTTC | 10–100 copies of template plasmid/reaction (raw wastewater) | [191] |

The LAMP method showed great potential in the application to wastewater samples because of its advantages such as inhibitor resistance, short reaction time (<1 h), and no need for advanced thermal cycling instruments [192]. By targeting the lmo0753 gene,

Nathaniel et al. developed a loop-mediated isothermal amplification assay for the detection of *L. monocytogenes* in wastewater. The LoD was 65 fg/µL of DNA and 38 CFU/mL, which was 10 times more sensitive than conventional PCR with primers targeting the HlyA gene. However, in the application to wastewater, a pre-culture procedure at 37 °C for 48 h was required [179]. An SA23 probe targeting *Salmonella* specifically by FISH has been developed by Santiago et al. (2008). The SA23 probe was shown to be capable of rapid and specific identification and visualization of *Salmonella* cells directly in the sample. By combining with the pre-enrichment, it could achieve the detection of 1 CFU/mL in seeded meat products [137]. This study also demonstrated the resistance of FISH to inhibitory substances in wastewater and the ability to differentiate viable but non-culturable (VBNC) cells. The advantage of the FISH method is that it is not inactivated by inhibitors and does not depend on the type of sample, even when dealing with large numbers of samples. Sequencing is also a powerful analysis tool for not only detection and relative quantification but also absolute quantification of bacteria in environmental samples, similar to the PCR-based method [193]. By spiking the samples with internal microorganism markers, the absolute bacterial number of targeting microbiomes could be calculated through the abundance of the internal markers with a known number. Different types of spiking markers have been used previously, including indigenous microorganism, synthetic, and heterogeneous markers [193–195]. The reference markers and spiking strategy should be optimized because only validated markers can be used to achieve reliable results [196].

In the application of biosensor-based methods, surface-enhanced Raman scattering/spectroscopy (SERS) has a high sensitivity, although its stability is unsatisfactory, and that limits its application in wastewater analysis [149]. Furthermore, colorimetric and fluorescent sensors generally have poorer LoDs than electrochemical, and they are easily disrupted by colored or turbid samples. Thus, it seems that the electrochemical aptasensor is more reliable in wastewater matrices [197]. In addition, the biosensor-based method showed good stability in wastewater samples. For example, Riquelme et al., developed a stable oligonucleotide-functionalized gold nanosensor for mecA ARG monitoring in 2017 [180]. This mecA-specific nanosensor can keep stable under environmental conditions and at high ionic strength, and it can demonstrate high selectivity even in the presence of target interference. This study supports the environmental suitability of a new, low-cost, field-deployable, and large-scale ARG analysis tool.

Most detection and quantification methods for pathogenic bacteria in wastewater involve sampling, which is followed by transportation to a central analytical laboratory for further analysis. In comparison, paper-based devices could achieve multiplexed, sensitive assays that rival PCR-based methods and provide high-quality, fast precision on-site diagnostics for pathogens. Although wastewater is a complex substrate, paper-based devices have shown the potential to detect pathogens in wastewater. Due to the outbreaks of COVID-19, paper-based devices are being quickly developed and employed in SARS-CoV-2 detection in sewage. In another previous study, it was successfully employed in the detection of various genes and microbiomes. A fully disposable and integrated paper-based sample-in-answer-out device was developed for nucleic acid testing, which can sensitively detect *S. typhimurium* with a detection limit of as low as $10^2$ CFU/mL in wastewater [181]. The presence of β-lactamase-mediated resistance was also detected using paper-based analytical devices (PADs). It was shown that, compared to traditional methods including culture methods, antibiotic susceptibility testing, and PCR gene analysis, their method can still reduce the laboratory processing time by 14–20 h, although a laboratory is still required to concentrate the wastewater samples [157].

### 4.3. Profiling Potential Pathogens

Sequencing, as a powerful analysis tool, has been widely used for profiling bacterial diversity and potential pathogens in wastewater. The DNA sequencing-based method can perform large-scale parallel analysis of PCR products and environmental nucleic acids. This provides a new dimension for the analysis of pathogenic bacteria in wastewater. The

application methodology of NGS technologies in wastewater study can be divided into four subcategories: whole genome sequencing (WGS), metagenomic sequencing, metatranscriptomic sequencing, and sequencing of an amplified targeted gene (e.g., 16S rRNA and 18S rRNA genes) [198]. WGS is a powerful approach for microorganism identification in wastewater, while it relies on bacteria isolation and culture, extraction of long DNA, and the development of long read sequencing platforms. Future development on these aspects could advance and simplify its application in wastewater analysis [199].

To date, sequencing assays based on the amplified gene regions take the dominant place in wastewater analysis. Microbial communities of waterborne pathogens were often studied by targeting high-variation region sequences of small subunit (SSU) rRNA genes (e.g., V1, V3, V4, V6) and large subunit (LSU) rRNA genes [182]. By using the full-length 16S rRNA gene sequence, Numberger et al. characterized and compared bacterial communities of the influent and effluent of a WWTP in Berlin, Germany [184]. The study found that during sewage treatment, the relative abundance of most pathogenic bacteria was effectively reduced, while *Legionella* and *Leptospirosis* showed an increase in relative proportion from inflow to effluent. This indicated that WWTPs may enrich and release certain potential pathogens into the environment, although they are effective in removing enteric bacteria. Oluseyi et al. studied the presence of pathogenic bacteria in three WWTPs in South Africa. Their study also confirmed the presence of bacterial pathogens in treated effluent, which may pose a potential contamination risk by transmission through soil, agriculture, water, or sediments [185]. Using Illumina MiSeq sequencing, Xue et al. (2019) investigated the spatial and temporal variability of bacterial structure and the presence of a human-associated Bacteroidale (HF183) marker in two WWTPs. Their findings illustrated how changes in bacterial communities can serve as a reliable means of monitoring the quality and performance of wastewater treatment plants for public and environmental health purposes [200].

Metagenomic study is an emerging methodology based on the sequencing data of genetic material recovered directly from environmental samples. This method has developed rapidly in the detection of microbial communities and their functional capabilities in wastewater. Currently, the application of metagenomics in wastewater is commonly employed for the identification of ARGs and genes associated with pathogens [183,201]. It is also increasingly utilized to support the assembly of whole or partial genomes from short-read sequencing data acquired from uncultured microbial communities [202]. Metatranscriptomic sequencing has similar principles with metagenomic sequencing, but it targets RNA rather than DNA, which is essential for identifying RNA viruses. Sequencing the transcribed mRNA could provide the information about which microbes are functionally active, as mRNA degradation rate varies among different species, thus challenging the preservation and analysis of mRNA [203]. In addition, the excess of ribosomal RNA (rRNA) in transcriptomes also interferes with the identification of mRNA in environmental samples. All these challenges limit its application in wastewater systems [198]. Comparison studies have also been conducted to access the performance of various methods to identify pathogens and associated virulence genes. For example, Yergeau et al. compared pre- and post-treatment biosolids from two WWTPs by using enumeration methods combined with molecular techniques including quantitative PCR, 16S rRNA and cpn60 gene amplicon sequencing, and shotgun metagenomic sequencing [204]. Their study showed that shotgun metagenomics indicted the widest range of pathogen DNA and was the only method that can obtain functional gene information in treated biosolids among all approaches.

### 4.4. Antimicrobial Resistance Analysis

With the growing concern about antimicrobial resistance, the WHO established the Global Antimicrobial Resistance Surveillance System (GLASS) in 2015 for sharing information on a global level to strengthen data on national and international actions and to aid decision making [205]. WWTPs treating wastewater from different sources provide a suitable circumstance for the emergence and spread of antibiotic resistance genes (ARGs)

and antibiotic resistant bacteria (ARB) [79,206]. Analysis of ARGs in influent wastewater can provide a broader perspective for the study of ARGs in the population. Various ARGs have been investigated and reported in wastewater based on qPCR technology [207,208]. A quantitative analysis of ARGs and horizontal gene transfer (HGT) potential was conducted over four seasons at a WWTP using a high-throughput qPCR array [166]. In this study, 285 primer sets targeting ARGs and nine transposase genes related to HGT were successfully used with wastewater samples. A microfluidic quantitative polymerase chain reaction (MF-qPCR) method was developed and optimized for simultaneously quantifying 16S rRNA genes, ARGs, heavy metal resistance genes, and an integrase gene that encodes three different types of integrons. This MF-qPCR method has better detection limits than shotgun metagenomics, which has also been used to detect large amounts of ARGs in other studies [92,209].

Sequencing-based methods also contributed to the analysis of ARGs and ARB (e.g., prevalence, relative abundance, and persistence) in wastewater. Using metagenomic analysis, a pipeline covered the analysis of gene transfer potential and the potential, pathway, and phylogenetic origin of ARGs was developed for identifying antibiotic resistance determinants in wastewater samples [99,182,183]. Meanwhile, discrepancies in ARG quantification by using different sequencing approaches have been reported by several previous studies where some ARGs were only identified by Nanopore sequencing and some others were only detected via Illumina sequencing [210]. Sequencing platform biases on the ARG quantification outputs were due to different ARG identification or prediction algorithms. The Illumina algorithm is based on the similarity search using BLAST, whereas Nanopore sequencing is based on workflows designed for the alignment of long high-error-rate sequences [211].

### 4.5. Prospect of Molecular Methods for Pathogenic Bacteria in Wastewater Analysis

Based on the above recent advances of various molecular methods, their advantages and limitations in wastewater analysis are summarized in Table 6. All molecular methods are able to achieve the detection of pathogenic bacteria in wastewater with an LoD range from 1 to 100 CFU or gene copies per 100 mL. However, the LoD varies according to different targeted pathogenic bacteria and sample pre-treatment procedures. Overall, the sensitivity of the current molecular methods is adequate for WBE purposes.

**Table 6.** Advantages, limitations, and prospects of various molecular methods in analyzing pathogenic bacteria for WBE applications.

| Molecular Method | Biomarkers | Advantages | Limitations | Reference |
|---|---|---|---|---|
| **Nucleic acid targeting methods** | DNA/RNA | - High sensitivity<br>- High specificity<br>- Multiple targets detection and quantification<br>- Fast community profiling | - Require sample storage and processing<br>- Require DNA/RNA extraction, which can cause DNA/RNA loss<br>- Sensitive to inhibitors<br>- High cost for large number of samples<br>- Usually need specialized instruments | [11,212] |
| **Immunology-based methods** | Proteins | - Low cost<br>- Can be automated<br>- Can detect bacterial toxins | - Require pre-enrichment<br>- Low sensitivity<br>- Require labeling of antibodies and antigens | [9] |
| **Biosensor-based methods** | DNA/RNA, proteins, chemicals | - High sensitivity<br>- Real-time detection<br>- Label free | - High cost<br>- Require specialized instruments<br>- Low specificity<br>- Not suitable for simultaneous detection of various organisms<br>- Low reproducibility and insufficient stability | [197] |
| **Paper-based device** | DNA/RNA, proteins, chemicals | - Cost effective<br>- Instrument free | - Detection limit (LoD) is usually high due to the traditional colorimetry<br>- Limitations of the structure and material of paper device | [13,154,156] |

Nucleic acid targeting methods have the potential to become the most suitable molecular method for wastewater analysis because one extracted nucleic acid sample could be

analyzed by various methods through different downstream workflows to achieve comprehensive analysis, including target gene detection and quantification (relative/absolute), microbiome community profiling, and ARG/functional gene analysis. Moreover, nucleic acid targeting methods could realize the direct identification and quantification of specific genes rather than gene expression products and thus could reduce the mistakes induced by some sample pre-treatment procedures, including enrichment and cultivation. The methods need to be improved to overcome limitations including sensitivity to inhibitors in wastewater, DNA/RNA loss caused by sample processing, and high cost. In addition, more specific gene biomarkers including species-specific genes, host-specific genes, and reliable cell quantification genes (one DNA/RNA fragments per cell) should be carefully selected. Standardization for sample storage, pre-processing, inner-extraction control, recovery methods, and certain analytical methods will highly expand their prospects in WBE study.

Immunology-based methods are powerful methods for detecting and capturing gene expression products such as cell surface proteins and bacterial toxins. For wastewater analysis, they were widely used in specific bacteria detection, isolation, and enrichment. They are more suitable for specific pathogen and microbial antigen detection and isolation from wastewater. They can also combine with other molecular methods to achieve deep analysis of a target pathogen's genome. By combining with a biosensor platform or paper-based device, they are expected to achieve fast, on-site, real-time, and low-cost identification and enumeration of pathogens in wastewater.

Biosensor-based methods have showed great potential in pathogenic bacteria analysis in wastewater. However, in the real wastewater analysis, they are not cost-effective compared to other molecular methods and cannot process large numbers of environmental samples, such as wastewater, from a long-term surveillance point-of-view. Paper-based devices are a good platform to be combined with many other molecular detection methods to achieve fast analysis of pathogenic bacteria in wastewater. Their application prospect towards wastewater has been well confirmed as a fast, on-site, cost-effective, portable, and disposal device. However, their sensitivity and specificity should be further improved towards detecting various targets in wastewater.

According to different research objectives, appropriate molecular methods can be selected and combined to achieve satisfactory analytical performance. The sensitivity and specificity of different molecular methods should be evaluated on various wastewater samples to improve their analytical performance. Infectious disease surveillance based on WBE could be divided into three main phases including pathogen monitoring in raw sewage, disinfection evaluation and ARG variation in WWTPs, and risk evaluation for further spreading of effluent in the environment. Among those three phases, nucleic acid targeting methods are more suitable for the pathogen monitoring of raw sewage since this phase requires the fast analysis of deep and comprehensive genomic information of pathogenic microbiomes delivered from communities. Especially for the sequencing-based method, with the decreasing cost and by combining with different workflows, it is expected to realize all kinds of analysis, including identification, relative and absolute quantification, bacterial community profiling, and ARG and functional gene analysis. This sets it apart from other nucleic acid targeting methods. For disinfection and ARG evaluation in WWTPs, except for nucleic acid targeting methods, immunology-based methods and biosensor-based methods are powerful analysis tools for the assessment of the activity and infectivity of pathogenic bacteria. For the risk evaluation of effluent and the environment, since the concentration of pathogenic bacteria in effluent and in the surrounding environment is usually low, and this evaluation does not require accurate detection and quantification results, the paper-based device was considered as the best choice since it can achieve fast, cost-effective, and on-site screening of the concentration level of target genes and pathogens. In addition, immunology-based methods could also be used in this phase to evaluate the infectivity and activity of pathogenic bacteria for risk assessment. However, the gold standard approach of sampling, transport and storage, pre-treatment, and the

enrichment of biomarkers in wastewater for various downstream molecular detections would dominate in WBE applications. The efficient recovery of high-quality biomarkers from wastewater samples should thus be the priority of future method development.

## 5. Conclusions

With the ever-increasing concerns over infectious diseases caused by pathogenic bacteria and their antimicrobial resistance, it is widely recognized that effective surveillance is key to the rapid intervention and control of outbreaks of infectious diseases. Wastewater-based epidemiology has become a popular tool due to its great potential as a population prevalence surveillance system and an early warning tool for disease outbreaks. With the development of molecular techniques for the detection of pathogenic bacteria and associated biomarkers, WBE applications are expanding to cover a wide range of pathogens using different advanced molecular methods tailored for wastewater analysis.

PCR-based methods have high sensitivity, and they are broadly used for the rapid analysis of pathogenic bacteria in wastewater, following DNA/RNA extraction procedures. Methods such as DNA microarray and sequencing-based methods are suitable for the in-depth study of bacterial communities and the presence of pathogenic bacteria and antimicrobial resistance due to their capability of large-scale parallel analysis of the whole microbiome. Alternative nucleic acid targeting methods such as FISH and LAMP are relatively sensitive, specific, and cost-efficient. However, nucleic acid targeting methods are not able to provide information about the activity and infectivity of various pathogens in wastewater. In addition, the need for sample pretreatment and research into multiplexing microorganism detection in a single sample are still challenges. Biosensors are easy to operate and do not need trained personnel for the detection of pathogenic bacteria in wastewater. Moreover, paper-based devices have recently emerged and have been widely used in pathogenic bacteria detection in wastewater because of their rapidness and cost effectiveness.

Molecular methods have found their diverse applications in WBE. Nucleic acid-based methods enable the direct and comprehensive analysis of the DNA/RNA of wastewater samples, including target gene (e.g., species-specific genes, ARGs, and functional genes) detection and quantification (both direct and relative quantification), profiling of the whole microbiome in the sample, genome sequencing, and analyzing. Thus, nucleic acid-based methods have the broadest prospect in wastewater analysis. Biosensor-based methods and paper-based devices exhibited great potential in fast and on-site detection of chemicals and microbiomes, which is suitable for the early warning of infectious disease outbreaks. Although, it seems that the application of immunology-based methods in wastewater is limited by a lot of disadvantages, it is the only method that targets the gene expression products of microbiomes. A lot of biosensors and paper-based devices are developed based on immunology-based methods. It is essential to improve immunology-based methods to suit WBE applications.

Although molecular methods have shown great potential in the analysis of pathogenic bacteria in wastewater, there are still several key challenges for their application in WBE. The low concentration of pathogenic bacteria in wastewater and inhibition of the complex wastewater matrix are additional concerns for these methods in comparison to the analysis of other types of samples. The low DNA/RNA recovery efficiency of pathogenic bacteria from wastewater needs to be improved and reported quantitatively with the results. The accuracy and reliability of WBE would be significantly enhanced with well-established molecular detection methods.

**Author Contributions:** Conceptualization, S.Z. and G.J.; writing—original draft preparation, S.Z.; writing—review, S.Z.; X.L.; J.W.; L.C.; J.O.; and F.H.; supervision, funding acquisition, and writing—review and editing, G.J. All authors have read and agreed to the published version of the manuscript.

**Funding:** This research was supported by the ARC Discovery project (DP190100385). Shuxin Zhang receives the support from a University of Wollongong PhD scholarship.

**Institutional Review Board Statement:** Not applicable.

**Informed Consent Statement:** Not applicable.

**Conflicts of Interest:** The authors declare no conflict of interest.

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
