# Peer review of "Molecular Methods for Pathogenic Bacteria Detection and Recent Advances in Wastewater Analysis"

_water, doi:10.3390/w13243551_

Round 1

Reviewer 1 Report

The manuscript entitled “Molecular methods for pathogenic bacteria detection and recent advances in wastewater-based epidemiology” reviews the application of molecular analysis in the detection of some pathogenic bacteria (E. coli O157:H7, Campylobacter, Legionella, etc.) and biomarkers (nucleic acids, proteins and antigens) in wastewater. These molecular analysis methods mainly include PCR-based methods, DNA Microarray, Fluorescent in situ hybridization (FISH), Paper-based methods and so on. The authors summarized the research progress of these methods, which have certain practical significance for the detection of pathogenic bacteria in wastewater. In general, the content of the manuscript is relatively substantial. But there are still some problems need to be solved in the manuscript, such as language, format, content accuracy and so on.

Some specific concerns and suggestion are as follows:

  1. It is important to note that the manuscript needs to be carefully edited by native English speakers, paying special attention to grammar, spelling and sentence structure so that the reader can clearly understand the objectives and results of the research.
  2. The title of the manuscriptcontains research advances in wastewater-based epidemiology, but it is not fully reflected in the content of the manuscript. Please revise it carefully.
  3. In the part of “3.2 molecular methods”, this manuscriptintroduces the detection methods based on nucleic acid more abundant, but other methods are not enough content.
  4. In the part of “4. Recent advances of molecular methods for pathogenic bacteria in wastewater”, why introduced the sample processing in manywords? This is not consistent with the content of this section.
  5. In the part of “4.2 Detection and quantification of specific pathogenic bacteria”, some methods only introduce their sensitivity and detection advantages, but not their specificity.
  6. Some references are too old and need to be updated.

Reviewer 2 Report

This review is well written. The authors know well the content.  It is easy to read in spite of the fact that this is long and there are many new details.  

There are only some modest proposals to correct.

Line 91: Pathogenic microorganisms in wastewater (not bacteria).

Line 214: Treatment of infections 

Line 506: False-negative results are a serious problem. When using classical methods this problem is also usual. The reasons can be inhibitory compounds,  and in addition, the content of wastewater, feces, sediments, etc. vary highly. It is possible that some subsamples are rich with interesting organisms while others are not.    

Table 6: It would be better to present this as text. Are there any temperature limitations for paper-based methods?

In the reference list, the reference 56 looks unusual. Check! See the instructions for authors and correct them as the name of journal and volume are italic and bold for the year.     

Reviewer 3 Report

Review points:

  1. Line 14-16- This sentence is not well connected with the earlier sentence. Why and how only function genes and ARG? function genes and ARG heavy terms might need more background on it in the previous sentence.
  2. Line 16-17- How do molecular methods replace the culture-based method? instead, you can say molecular methods are gradually emerging…
  3. Line 18-19- Again new topic, has not connected with the previous sentence. Why you do not start with WBE- and connect with the current molecular approach
  4. Overall, abstract: now sentences are highly fragmented, need to create a story.

Introduction

  1. Line 28-29- Rewrite this sentence- You can choose one of these words: Infectious disease- gastrointestinal disease- systematic illness - my opinion all are not needed.
  2. Again the second sentence is not connected with the first-
  3. Line 50- first give the advantages of the culture-based method. It is also an important tool in public health microbiology- Good reference: https://doi.org/10.3390/ijerph18115513, (Pond, 2006; Ananda Tiwari, Oliver, et al., 2021)
  4. Line 51- what are target/endogenous?
  5. line 54- Underestimate target count but it is the way how it works. Instead, you can say VBNC can become viable and may cause disease and increase public health risk (Ananda Tiwari, Oliver, et al., 2021).
  6. Line 62: digital PCR, not droplet digital PCR, see ddPCR is a type of dPCR (Mao et al., 2019).  
  7. Line 96- give reference. potential references(Agency, 2009; Pond, 2006; Who, 2003)
  8. Chapter 2.1- Take in your mind that you are writing a manuscript about infectious disease, wastewater-based surveillance. Zoonotic pathogens may not good for WBE. Or it needs a different way of interpretation of the result. You can add enteric viruses to this list. Viruses are highly hosted specific so finding them in sewage infer it is from the human population residing in the service area of WWTP. But your topic is pathogenic bacteria, which may be virus not relevant. At least you need to mention, bacteria can have diverse sources for example zoonotic, or environmental eg Legionella, Mycobacterium.
  9. Line 277- do not use words incorrect instead you can use false-positive/false-negative types of word
  10. Line 322-360- do you mean multiplex assay, what do you mean mPCR? PCR/qPCR/dPCR   are method. mPCR can be different from them. multiplexing is possible in qPCR but is more sensitive with dPCR. TaqMam/SYBR Green these are different, they use for reading the amplification- do not mix all together- create a story.
  11. Line- 367-68: need reference and it may not be true always, if there is inhibition, a nucleic acid may not amplify. It also has some limitations. Mention them too.
  12. Line 380- mRNA ? it can be low in environmental sample. Instead rRNA, (Inkinen et al., 2021; Pitkänen T, Ryu H, Elk M, Hokajärvi A-M, Siponen S, Vepsäläinen A, Räsänen P, 2013; A. Tiwari et al., 2016)
  13. Table 5- I think it does not need.

Overall: You have plenty of information. All the time, you have not bound the information on the thread of your objective. Think you are making a flower necklace, the aim of your paper is a thread of necklace and your information is flowers.

WBE- you can highlight, clinical detection is only for clinically sick people, but WBE can also be detected from asymptomatic, presymptomatic individuals who shed pathogen to them (Ahmed et al., 2021; Hokajärvi et al., 2021; Ananda Tiwari, Lipponen, et al., 2021).

Reduce irrelevant information on your topic.

Before submitting the updated version, circulate once the manuscript with your senior co-authors and approve it from them. Best of luck.

References

Agency, U. S. E. P. (2009). Review of Zoonotic Pathogens in Ambient Waters (Issue February 2009).

Ahmed, W., Bibby, K., D’Aoust, P. M., Delatolla, R., Gerba, C. P., Haas, C. N., Hamilton, K. A., Hewitt, J., Julian, T. R., Kaya, D., Monis, P., Moulin, L., Naughton, C., Noble, R. T., Shrestha, A., Tiwari, A., Simpson, S. L., Wurtzer, S., & Bivins, A. (2021). Differentiating between the possibility and probability of SARS-CoV-2 transmission associated with wastewater: empirical evidence is needed to substantiate risk. FEMS Microbes, March, 1–4. https://doi.org/10.1093/femsmc/xtab007

Hokajärvi, A. M., Rytkönen, A., Tiwari, A., Kauppinen, A., Oikarinen, S., Lehto, K. M., Kankaanpää, A., Gunnar, T., Al-Hello, H., Blomqvist, S., Miettinen, I. T., Savolainen-Kopra, C., & Pitkänen, T. (2021). The detection and stability of the SARS-CoV-2 RNA biomarkers in wastewater influent in Helsinki, Finland. Science of the Total Environment, 770. https://doi.org/10.1016/j.scitotenv.2021.145274

Inkinen, J., Siponen, S., Jayaprakash, B., Tiwari, A., Hokajärvi, A.-M., Pursiainen, A., Ikonen, J., Kauppinen, A., Miettinen, I. T., Paananen, J., Torvinen, E., Kolehmainen, M., & Pitkänen, T. (2021). Diverse and active archaea communities occur in non-disinfected drinking water systems–Less activity revealed in disinfected and hot water systems. Water Research X, 12, 100101. https://doi.org/10.1016/j.wroa.2021.100101

Mao, X., Liu, C., Tong, H., Chen, Y., & Liu, K. (2019). Principles of digital PCR and its applications in current obstetrical and gynecological diseases. American Journal of Translational Research, 11(12), 7209–7222.

Pitkänen T, Ryu H, Elk M, Hokajärvi A-M, Siponen S, Vepsäläinen A, Räsänen P, S. D. J. (2013). Detection of fecal bacteria and source tracking identifiers in environmental waters using rRNA-based RT-qPCR and rDNA-based qPCR assays. Environ. Sci. Technol., 47, 13611–13620.

Pond, K. & W. H. O. (2006). Water Recreation and Disease – Plausibility of Associated Infections: Acute Effects, Sequelae and Mortality. In World Health Organization. https://apps.who.int/iris/handle/10665/43338 (Vol. 19, Issue 2). https://doi.org/10.1108/ijhcqa.2006.06219bae.001

Tiwari, A., Niemelä, S. I., Vepsäläinen, A., Rapala, J., Kalso, S., & Pitkänen, T. (2016). Comparison of Colilert-18 with miniaturised most probable number method for monitoring of Escherichia coli in bathing water. Journal of Water and Health, 14(1). https://doi.org/10.2166/wh.2015.071

Tiwari, Ananda, Lipponen, A., Hokajärvi, A., & Luomala, O. (2021). Detection and quantification of SARS-CoV-2 RNA in wastewater influent in relation to reported COVID-19 incidence in Finland. MedRxiv Preprint, 1–37. https://doi.org/10.1101/2021.10.05.21264462

Tiwari, Ananda, Oliver, D. M., Bivins, A., Sherchan, S. P., & Pitkänen, T. (2021). Bathing Water Quality Monitoring Practices in Europe and the United States. International Journal of Environmental Research and Public Health, 18(11), 5513. https://doi.org/10.3390/ijerph18115513

Who. (2003). Faecal pollution and water quality. Guidelines for Safe Recreational Environments. Volume 1: Coastal and Fresh Waters, 51–101. http://www.who.int/water_sanitation_health/bathing/srwe1-chap4.pdf

Round 2

Reviewer 3 Report

Now it has done a lot. It is quite great now. Regarding the topic” Molecular methods for pathogenic bacteria detection and recent advances in wastewater analysis"

Can it be better (just my proposal/opinion) not necessarily you need to follow me-

Does this better represent your paper?

"Molecular methods for pathogenic bacteria detection and recent advances for environmental surveillance"